# `GCFed`: Exploiting Gradient Correlation for Client Selection and Rate Allocation in Federated Learning

## Abstract

Federated Learning (FL) has gained increasing popularity for its ability to harness diverse datasets from multiple sources without the need for data centralization. Extensive research has focused on reducing the cost of communications between remote clients and the parameter server. However, existing works fail to comprehensively leverage the correlation among the gradients at the remote clients. In this work, we propose `GCFed` – a novel FL framework that exploits the clients' gradient correlation to reduce communication costs while maintaining satisfactory convergence. Specifically, we propose an information-theoretic problem formulation that considers the model update problem in a single FL iteration as a multi-terminal source coding problem in the context of rate-distortion theory. We solve the associated optimization problem using convex semidefinite relaxation techniques with an iterative algorithm and leverage the solution to develop a joint approach for correlation-aware client selection and rate allocation. Extensive experiments are conducted to validate the effectiveness of our proposed framework and approaches as compared to state-of-the-art methods. Our code is available at: `https://anonymous.4open.science/r/GCFed-D03B`.

## 1 Introduction

Among the various distributed deep learning paradigms developed in the literature, Federated Learning (FL) has garnered significant attention for its ability to train machine learning models on decentralized data. However, FL faces several challenges, including data heterogeneity, client availability, and communication efficiency. Among these challenges, communication efficiency is perhaps the most critical, as FL often requires frequent model exchanges between the remote clients and the Parameter Server (PS). Therefore, the focus of our investigation is reducing communication costs in FL while ensuring fast model convergence.

Significant efforts have been made in the literature to both (i) theoretically characterize the relationship between communication efficiency and model convergence, and (ii) develop efficient algorithms to reduce communication overhead in FL scenarios of practical relevance. Most existing approaches envision a dynamic scenario in which clients continually join and leave the training process. This implicitly leads to problem formulations, theoretical analyses, and implementations solely based on the properties of local gradients. These solutions are generally rather effective in reducing the overall communication costs between clients and the PS. While scalability is important for FL applications that envision a vast amount of clients – as in the original vision of McMahan & Ramage (2017) – we believe that scenarios with fewer clients will benefit from further coordination between the PS and each remote client. For instance, in the case of a data-center scale learning instances, one could envision sacrificing scalability to obtain a further reduction in communication. This is the scenario explored in this paper. More specifically we leverage the gradient correlation across clients to improve the communication efficiency in FL. From a high-level perspective, we seek to answer the following questions: "How to enable the PS to collect statistics on the correlation among the clients' local gradients, and how this information can be best utilized to reduce the communication overhead?"

We answer the questions through an information-theoretic formulation, and practical client selection and rate allocation implementations based on it. We take inspiration from Zhang et al. (2023); Yang et al. (2023),

where the authors propose to model the single iteration communication problem with a CEO problem Berger et al. (1996). The CEO problem is an information-theoretic model where a set of remote clients observe the same source through different noisy channels, with the goal of reconstructing it at a central terminal under a mean square error (MSE) distortion constraint. The rate-distortion function for this model corresponds to the ultimate compression performance that can be attained in the regime of infinite source observations and under no complexity considerations. The formulation of Zhang et al. (2023); Yang et al. (2023) provides a compelling perspective on the problem of communication efficiency in FL, but their results are limited to scenarios where the clients' local datasets are homogeneous. Inspired by their insights, we adopt a variant of the CEO problem formulation with correlated local gradients and provide technical results applicable in heterogeneous FL.

In this paper, we propose `GCFed` (Gradient Correlation-aware Federated Learning), the first framework that exploits the clients' gradient correlation to improve communication efficiency and convergence. Our contributions can be outlined as follows:

- **Problem Formulation:** We propose an information-theoretic formulation that frames the model update problem in a single FL iteration as a multi-terminal source coding problem, viewed from a sum-rate-distortion perspective.

- **Technical Results:** We bound the minimum sum-rate from above and below via convex semidefinite relaxation and propose an algorithm that efficiently evaluates the bounds by computing the saddle point of a max-min problem.

- **Practical Implementations:** Building upon our theoretical results, we develop a joint approach that involves layer-wise client selection and rate allocation using an estimate of the gradient covariance matrix obtained by subsampling the clients' local gradients.

- **Numerical Evaluations:** We conduct extensive experiments and compare with the state-of-the-art approaches to demonstrate the effectiveness of our proposed methods.

## 2 Related Work

**Theoretical analysis**. Various authors in the literature have studied the trade-off between communication cost and convergence in FL. Early works characterize the global model convergence rate by adopting the assumption that the local datasets are uniformly distributed Zhou & Cong (2018); Stich (2019); Wang & Joshi (2019) or heterogeneously distributed Li et al. (2020). In these studies, the channel between the clients and PS is assumed to be noiseless but rate-limited. More recent literature has considered a different communication constraint in which the total number of clients participating in the training process is bounded. Accordingly, the PS must select which client should participate in a given FL round. Tan et al. (2022); Jee Cho et al. (2022) analyze the convergence of FedAvg under partial client participation. Rodio et al. (2023) considers the temporal and spatial correlations in clients' availability dynamics. Ribero et al. (2023) assesses intermittent client availability and improves FL convergence by learning clients' long-term participation rates.

The above perspectives of communication-constrained FL– either considering (i) a finite rate channel between the clients and the PS, or (ii) a finite set of active clients – can be unified under a single framework in which a constraint is posed on the total transmission between clients and PS. This rate can then be allocated equally to all clients or among a set of selected clients. The theoretical framework that allows one to analyze these two scenarios – and all those in between— is provided by the distributed source coding problem in the context of rate-distortion theory. The authors of Zhang et al. (2023); Yang et al. (2023) first propose formulating the model update problem in a single FL iteration as a distributed source coding problem. Yang et al. (2023) puts forth a framework for model aggregation performance analysis and derives the inner bound for the rate region. Zhang et al. (2023) first connects the model update problem with the classical indirect Gaussian multi-terminal source coding, also known as the CEO problem Berger et al. (1996); Oohama (1998). Unfortunately, their results are constrained to uniform data settings due to their assumptions.

**Practical Implementation**. Among practical implementations aimed at reducing FL communication costs, model compression and client selection approaches demonstrate promising results. Common techniques em-

ployed in model/gradient compression include sparsification Aji & Heafield (2017); Ivkin et al. (2019) and quantization Alistarh et al. (2017); Wen et al. (2017); Gandikota et al. (2021); Liu et al. (2023). Although communication efficiency is significantly improved by either applying such techniques separately or jointly, most existing literature neglects gradient correlation in rate allocation, leading each client to perform compression independently, which limits their potential.

For client selection, there are proposals to assign participating clients with dynamic weights for clients to mitigate aggregation bias in partial client schemes, with Rodio et al. (2023) considering temporal and spatial client correlation, while Tan et al. (2022) not. Ribero et al. (2023) proposes two algorithms for positively correlated and uncorrelated clients based on clients availability. Jee Cho et al. (2022); Goetz et al. (2019) develop biased client selection strategies favoring clients with higher local losses. Tang et al. (2022) employs Gaussian Process modeling to collectively analyze and leverage the correlation of local losses for client selection. Xu et al. (2024) introduces a heterogeneity-aware approach promoting diversity through top$K$ gradient sparsification and client selection strategies.

To the best of our knowledge, the correlation between gradients has not been fully exploited in existing FL literature.

## 3 Problem Formulation

### 3.1 Notations

Lowercase boldface letters (e.g., $\mathbf{z}$) are used for vectors, uppercase letters for random variables (e.g., $X$), and calligraphic uppercase symbols for sets (e.g., $\mathcal{A}$). Given the set $\mathcal{A}$, $|\mathcal{A}|$ indicates the cardinality of the set. We also adopt the short-hands $[m:n] \triangleq \{m, \ldots, n\}$ and $[n] \triangleq \{1, \ldots, n\}$. Let $M_1$ and $M_2$ be two square matrices of size $\mathbb{R}^{m \times m}$. The notation $\mathrm{diag}(M_1, M_2)$ represents the diagonal operator that places $M_1$ and $M_2$ on the diagonal of a matrix of size $2m \times 2m$, with the remaining entries zero-padded. We use "$\preceq$" to denote positive semidefinite (PSD) partial ordering.

### 3.2 Federated Learning Preliminaries

Consider the FL setting with $K$ clients, each possessing a local dataset $\mathcal{D}_k \in \mathcal{D}$, for $k \in [K]$ wishing to minimize the *loss function* $\mathcal{L}$ as evaluated across all the clients and over the model weights $\mathbf{w} \in \mathbb{R}^m$, where $m$ denotes the dimensionality of the model parameter. This minimization is coordinated by the PS as follows: in round $t \in [T]$, the clients transmit local gradients to the PS; the PS aggregates, generates a model update, and sends the updated model back to the clients. The above steps are repeated for $T$ times: the model obtained at time $T$ is declared as the converged model. Mathematically, the loss function $\mathcal{L}$ is defined as

$$\mathcal{L}(\mathbf{w}) = \frac{1}{|\mathcal{D}|} \sum_{k \in [K]} \mathcal{L}_k(\mathcal{D}_k, \mathbf{w}), \tag{1}$$

where $\mathcal{L}_k(\mathcal{D}_k, \mathbf{w})$ is the local loss function quantifying the prediction error of the $k$-th client's model. A common approach for numerically finding the optimal value of $\mathbf{w}$ is through the iterative application of (synchronous) stochastic gradient descent (SGD). We define the local gradients calculated at communication round $t$ as

$$\mathbb{E}[g_{kt}] = \mathbb{E}[\nabla \mathcal{L}_k(\mathcal{D}_k, \mathbf{w}_t)], \tag{2}$$

where $\nabla \mathcal{L}_k(\mathcal{D}_k, \mathbf{w}_t)$ denotes the local gradients of the model evaluated at the local dataset of the $k$-th client by minimizing the local loss function. Note that the expectation in equation 2 is taken over the randomness in evaluating the gradients, e.g., mini-batch effects.

In practical FL scenarios, the communication between clients and the PS is often limited. One common constraint considered in the literature on communication between clients and the PS is the *rate constraint*– in which the communication is restricted to $d\mathsf{R}_k$ bits from each client to the PS. To meet this constraint, each client $k$ compresses the $d$-dimensional gradient vector $\mathrm{g}_{kt}$ to a $d\mathsf{R}_k$ binary vector through a compressor $\mathrm{comp}_{\mathsf{R}_k} : \mathbb{R}^d \to [2^{d\mathsf{R}_k}]$, where the sum over all $\mathsf{R}_k$ is constrained, and the client is dropped (unselected) if

assigned a rate of $R_k=0$. The PS aggregates all the compressed gradients and forms the new global weights

$$\mathbf{w}_{t+1} = \mathbf{w}_t - \eta_t \hat{\mathbf{g}}_t, \text{ for } t \in [T], \text{ where} \tag{3}$$

$$\hat{\mathbf{g}}_t = \frac{1}{K} \sum_{k \in [K]} \text{comp}_{R_k}^{-1}(\text{comp}_{R_k}(\mathbf{g}_{kt})). \tag{4}$$

$\mathbf{w}_0$ is a random initialization. The ultimate goal in a single FL iteration is to minimize the distortion in gradient reconstruction, i.e., the distortion between $\mathbf{g}_t$ and $\hat{\mathbf{g}}_t$. We assume that minimizing the gradient reconstruction distortion leads to minimum perturbation of the accuracy.

### 3.3 Relevant Assumptions

In general, the iterations in equation 3 for the learning problem in equation 1 give rise to a random process when accounting for (i) the heterogeneity in the data distribution, (ii) the randomness in the network initialization and (iii) the randomness in the gradient evaluation. The characterization of the properties of this process is generally considered a hard problem. In the following, we adopt a set of assumptions on the distribution of the gradients that make it possible to develop a rigorous theoretical and numerical approach. Although some assumptions are not yet validated, the numerical results show the effectiveness of the approach developed from them. These gradient assumptions are as follows:

A1 **well-defined random process:** Given a learning problem as in equation 3, considering the randomness gives rise to the well-defined random process $\mathcal{G} = \{\mathbf{g}_{kt}\}_{k \in [K], t \in [T]}$.

A2 **independence over time:** We assume that the process $\mathcal{G}$ is such that $\mathbf{g}_{kt} \perp \mathbf{g}_{kt'}$ for $t \neq t'$.

A3 **iid-ness intra-layer:** For each client, the gradient is an i.i.d. sequence within each layer.

A4 **joint Gaussianity across clients:** For the same entry in the same layer, the distribution of gradients at client $k$ and client $k'$ follows a joint Gaussian distribution with mean zero and covariance $\Sigma_{k,k'}$. $\Sigma_{k,k'}$ is the $(k, k')$-entry of a covariance matrix $\Sigma$.

A5 **independence inter-layer:** For simplicity, the gradients across different layers are assumed to be independent in our problem formulation. One may view this as equivalent to considering a single-layer network theoretically. In Sec.4 and Sec.5, technical analyses and implementations are done layer-wise for multi-layer DNNs. A multi-layer generalization is discussed in App.A.4.

*Remark* 3.1. [**Change of Notation**] Based on the above assumptions, we model each layer of the gradient at client k as a single random variable with each element in the sequence being a different realization. We change the notation of the gradient sequence of length n at client k at round t to be $\{G_{kt}\}_{i=1}^{n}$, abbreviated as $G_k^n$ (omitting round t). In App.A.4, a random vector generalization is included.

### 3.4 Multi-terminal Source Coding Problem in FL

Given the above, we introduce a problem formulation that considers the model update issue in a single FL iteration as the direct Gaussian multi-terminal source coding problem.

As depicted in Fig.1, at communication round $t$, client1 through client$K$ each obtains a local gradient sequence of length $n$, denoted as $G_1^n, G_2^n, ..., G_K^n$. Let the source be a Gaussian source with mean zero and positive definite covariance matrix $\Sigma_G$. Following assumptions A3 and A4 that assume i.i.d.-ness within each local gradients sequence and correlation across different clients at the same-position-entry of local gradients, we let $G_k(i), i \in [n]$ denote the entries of local gradient sequence of the $k$-th client, $k \in [K]$. Each client encodes the local gradients using encoding functions $f_k: \mathbb{R}^n \to \{1, 2, ..., M_k\}$, $k \in [K]$ and sends the encoded gradients to PS at respective rates $R_k, k \in [K]$.

Differing from traditional distributed learning problems that focus on reconstructing each individual source, the decoder $\Phi$ in FL aims to reconstruct an aggregation function of the sources. Denote the FL reconstruction

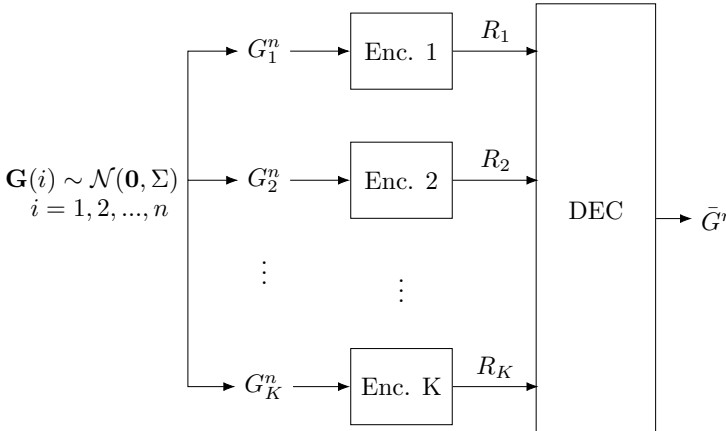

Figure 1: The Gaussian multi-terminal source coding problem in FL. See Remark 3.1 for notations.

as $\bar{G}^n$ with the same length as each source. Let $d$ denote the desired distortion level. The distortion constraint in FL is defined as

$$\frac{1}{n} \sum_{i \in [n]} \Big( \frac{1}{K} \sum_{k \in [K]} G_k(i) - \bar{G}(i) \Big)^2 \le d. \tag{5}$$

Here, we take FedAvg aggregation with an equal amount of local data as an example for simplicity. Note that our results could be generalized to any linear aggregation function with slight adjustments, as described in App.A.4.

Now we define the model update problem in a single FL iteration as the infimum of all achievable sum-rates for all possible encoders and decoder choices satisfying equation 5. The ultimate formulation of the problem we aim to solve is as

$$R^*(d) \triangleq \lim_{n \to \infty} \inf_{f_1, f_2, \ldots, f_K, \Phi} \frac{1}{n} \sum_{k \in [K]} \log M_k$$

$$\text{s.t.} \quad \frac{1}{n} \sum_{i \in [n]} \Big( \frac{1}{K} \sum_{k \in [K]} G_k(i) - \bar{G}(i) \Big)^2 \le d. \tag{6}$$

## 4 Proposed Methods

### 4.1 Overall Framework

In Algorithm 1, we present the overall framework. In `GCFed`, we assume that the remote clients employ the communication resources to transmit high-quality gradient samples. These high-quality samples are used to compute the covariance matrices, serving as the estimated correlation, which determines the set of selected clients $\mathcal{K}$ and the rate for each client $\mathcal{R}_k$ (a set due to layer-wise allocation). As our client selection and rate allocation schemes rely critically on the correlation between the clients' local gradients, we introduce a simple yet effective method to estimate the gradient correlation by random subsampling (see line 3-10 in Algorithm 1). The specific steps include:

- [**line 3**] PS randomly generates sets of indexes layer-wise and broadcasts them along with the global model to clients at the beginning of each communication iteration.

- [**line 5,6**] After finishing local updates, clients subsample their resulting gradients according to the indexes layer-wise and send the subsamples back to the PS.

- [**line 10**] Because all the clients sample from corresponding indexes layer-wise, the PS calculates one covariance matrix denoted as $\Sigma_G$, serving as the gradients correlation estimation across clients, for each layer.

---

**Algorithm 1** Training procedure of `GCFed`

---

1: Initialize global model $w_0$
2: **for** communication iteration $t = 1$ to $T$ **do**
3:     PS randomly samples index vectors $\vec{idx}$ for each layer of the model and broadcasts: $w_t$ and $\vec{idx}$
4:     **for** client $k = 1$ to $K$ in parallel **do**
5:         $G_{kt}^n \leftarrow$ **ClientUpdate**$(\mathcal{D}_k, w_t)$
6:         $G_{kt}^{(s)} \leftarrow$ **Subsample**$(G_{kt}^n, \vec{idx})$
7:         Client sends $G_{kt}^{(s)}$ back to PS
8:     **end for**
9:     PS performs:
10:         $\Sigma_G \leftarrow$ **CorrelationEstimation**$(G_{kt}^{(s)})$
11:         $\mathcal{K}_t \leftarrow$ **ClientSelection**$(\Sigma_G)$ , $\mathcal{K}_t \in K$
12:         $\mathcal{R}_{kt} \leftarrow$ **RateAllocation**$(\Sigma_G, \mathcal{K}_t)$
13:     PS broadcasts $\mathcal{K}_t$ and $\mathcal{R}_{kt}$
14:     **for** client $\tilde{k} = 1$ to $\mathcal{K}_t$ in parallel **do**
15:         $\hat{G}_{\tilde{k}t}^n \leftarrow$ **ClientEncode**$(G_{\tilde{k}t}^n, \mathcal{R}_{kt})$
16:         Client sends $\hat{G}_{\tilde{k}t}^n$ back to PS
17:     **end for**
18:     $w_{t+1} \leftarrow$ FedAvg$(\mathcal{K}_t,$ **ServerDecode**$(\hat{G}_{\tilde{k}t}^n, \mathcal{R}_{kt}))$
19: **end for**

---

Note that **ClientSelection**, **RateAllocation** and **ClientEncode** are performed in the same layer-wise manner as estimating the gradients correlation. **ClientUpdate** is chosen from deep learning optimization methods, for example, SGD. While incorporating correlation estimation into the FL framework introduces additional communication and computation costs, we emphasize that the costs incurred by our method are minimal compared to the benefits it provides.

### 4.2 Correlation-aware Client Selection Strategy

Given the number of clients to be selected, $n$, and the input covariance matrix resulting from the correlation estimation, $\Sigma_G \in \mathbb{R}^{K \times K}$, the client selection procedures are:

1. First, we construct a "combination matrix" $C \in \mathbb{R}^{m \times K}$, where $m = \binom{n}{K}$. Each row in $C$ is a boolean vector, representing one combination of choosing $n$ out of $K$ total clients; e.g., $[1, 0, 1]$ represents choosing the first and the third client out of three clients.

2. Then, we stack an all-ones vector at the bottom of $C$ and multiply the covariance matrix $\Sigma_G$ by $C$. We obtain a square matrix $Q$ of size $(m + 1) \times (m + 1)$, where $Q \triangleq C\Sigma_G C^T$. We note that the first $m$ diagonal terms of $Q$ represent the variances of the corresponding selection, and the last diagonal term represents the variance of considering all clients, namely, the variance of the true gradients.

3. Finally, by taking the difference between the last diagonal term of $Q$ and each of the first $m$ diagonal terms, we calculate a list of distortion values, with the smallest one corresponds to the optimal client selection.

The resulting client selection strategy yields the smallest expected distortion using (i) the gradients accumulated from the selected clients and (ii) all clients in the context of FedAvg, as compared to all other choices of active clients based on the covariance matrix $\Sigma_G$.

Note that this client selection strategy takes the form of matrix multiplication, which may leverage the GPU speed-up when dealing with a large number of clients: see App.A.1 for more details. We also highlight that all computations are performed at the PS, which is generally assumed to have substantial computational

---

**Algorithm 2** The iterative algorithm to evaluate $R^*(d)$

---

1: **Inputs:** $\Sigma_G$ and $d$
2: Initialize initial guesses for $\Sigma_1, ..., \Sigma_K$
3: **while** gap $\neq 0$ **do**
4:    Fix $\Sigma_1, ..., \Sigma_K$ and solve for the min part
5:    Obtain $D, \Gamma_1, ..., \Gamma_K$ and the lower bound $\underline{R}^*(d)$
6:    **while** max part is not converged **do**
7:       Fix $D, \Gamma_1, ..., \Gamma_K$ and linearize the objective function of max part at current $\Sigma_1, ..., \Sigma_K$
8:       Solve for the linearized max part
9:       Update $\Sigma_1, ..., \Sigma_K$ and the upper bound $\bar{R}^*(d)$
10:   **end while**
11:   Compute the gap between $\underline{R}^*(d)$ and $\bar{R}^*(d)$
12: **end while**

---

capability. The proposal to conduct client selection layer-wise is new in the FL literature. Its effectiveness is verified and explained in Sec.5.

## 4.3 Optimal Rate Allocation

At each communication round, given a covariance matrix $\Sigma_G$ and a desired distortion level $d$, we aim to calculate the minimum sum-rate as defined in equation 6. Explicit computation of equation 6 is challenging; therefore, we draw inspiration from Wang et al. (2010); Wang & Chen (2014) to establish its upper and lower bounds. To this end, we introduce several auxiliary variables (more details in App.A.2).

Based on any given $\Sigma_G$, we define a positive definite covariance matrix $\Sigma_Z$ such that $\Sigma_G^{-1} + \Sigma_Z^{-1}$ is a diagonal matrix, with diagonal elements $\Sigma_k^{-1}, k \in [K]$. Then we define $D$ as the distortion matrix of the minimum mean square error (MMSE) estimation for all the sources given all encoders' outputs and a diagonal matrix with diagonal elements $\Gamma_k^{-1}, k \in [K]$ representing the MMSE estimation of each source based on the corresponding encoder's outputs. With these auxiliary variables, the Berger-Tung upper bound Berger (1978); Tung (1978) for the minimum sum-rate defined in equation 6 is given by the following:

$$\bar{R}^*(d) \triangleq$$

$$\max_{\Sigma_1,...,\Sigma_K} \min_{D,\Gamma_1,...,\Gamma_K} \frac{1}{2} \log \frac{|\Sigma_G + \Sigma_Z| \, |\mathrm{diag}(\Sigma_1,...,\Sigma_K)|}{|D + \Sigma_Z| \, |\mathrm{diag}(\Gamma_1,...,\Gamma_K)|}$$

$$\text{s.t.} \quad \Sigma_Z = (\mathrm{diag}(\Sigma_1,...,\Sigma_K)^{-1} - \Sigma_G^{-1})^{-1},$$

$$0 \preceq \mathrm{diag}(\Sigma_1,...,\Sigma_K) \preceq \Sigma_G,$$

$$\mathbb{1} \cdot D \cdot \mathbb{1}^T \leq d,$$

$$0 \preceq \mathrm{diag}(\Gamma_1,...,\Gamma_K) \preceq \mathrm{diag}(\Sigma_1,...,\Sigma_K),$$

$$\mathrm{diag}(\Gamma_1,...,\Gamma_K) = (D^{-1} + \Sigma_Z^{-1})^{-1}. \tag{7}$$

Note that the maximization part of equation 7 is concave, but the minimization part is not convex. Therefore, we perform convex semidefinite relaxation on the last constraint by replacing "=" with "$\preceq$" (see equation 21) to make the minimization part efficiently solvable. It should be emphasized that after this relaxation, we not only obtain a lower bound for the Berger-Tung upper bound but also a lower bound for $R^*(d)$. Specifically, as shown in Algorithm 2, we develop an efficient iterative algorithm to solve the relaxed problem, which yields a lower bound for $R^*(d)$; substituting the optimal $\Gamma_1, \ldots, \Gamma_K$ back gives rise to an upper bound for $R^*(d)$. The algorithm computes the saddle point of the relaxed optimization problem by iteratively solve for the min part and the max part until two bounds coincide. Details are presented in App.A.3 due to page limits. Ultimately, the convex semidefinite relaxation contributes in two aspects: transforming the problem into a convex form that can be solved efficiently and allowing for the simultaneous calculation of both upper and lower bounds for $R^*(d)$.

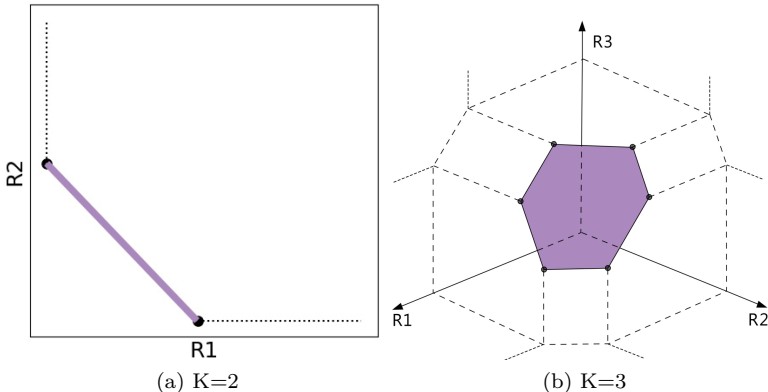

(a) K=2                    (b) K=3

Figure 2: Example dominant faces (colored) that describe the optimal rate allocation schemes.

Implicitly, characterizing the rate-distortion function also characterizes the set of rates attaining $\bar{R}^*(d)$: this solution corresponds to the optimal rate allocation at each client. Obtaining the results of $\Sigma_k$ and $\Gamma_k$ from equation 7, we define $M_k$, for $k \in [K]$ to be a diagonal matrix as

$$M_k = \text{diag}(\Sigma_1, ..., \Sigma_k, \Gamma_{k+1}, ...\Gamma_K). \tag{8}$$

The optimal rate allocation for each client $k \in [K]$ follows

$$R_k = \frac{1}{2} \log \frac{|\Sigma_Z(\Sigma_Z - M_k)^{-1}\Sigma_Z|}{|\Sigma_Z(\Sigma_Z - M_{k-1})^{-1}\Sigma_Z|} + \frac{1}{2} \log \frac{|\Sigma_k|}{|\Gamma_k|}, \tag{9}$$

satisfying the sum-rate constraint: $\bar{R}^*(d) = \sum_{k \in [K]} R_k$.

**Insights:** Contrary to the correlation-aware optimal rate allocation described in equation 9, there exists a correlation-agnostic version of the optimal rate allocation that also satisfies the same distortion constraint (see App.A.5). We numerically compare the two counterparts in Sec.5.4 and Sec.5.5, involving simulating the ideal quantization process. The difficulty in implementing the ideal quantizers lies in the fact that each remote client only has access to its own gradient, leading each client to design its quantizer by accounting for the expected reconstructing power that PS obtains when collecting all compressed gradients. From a purely information-theoretic standpoint, this can be attained through *binning* [1]. In practice, such implementations are not yet mature.

Another noteworthy feature of our proposed approach is that there indeed exist an infinite number of optimal rate allocation schemes attaining the same optimal sum-rate. The region formed by the equivalent rate allocations is depicted in Fig.2 for $K = 2$ (left) and $K = 3$ (right). This region corresponds to the dominant face of a polymatroid arising from the inequalities in equation 7, and the corner points are associated with a specific decoding order of the client, giving rise to $K!$ corner points, which can be calculated using variants of equation 9. More detailed insights are included in App.A.5.

## 5 Experiments

### 5.1 Simulation Settings

Experiments are conducted on the CIFAR-10 Krizhevsky et al. (2009) and Fashion-MNIST datasets Xiao et al. (2017) with a VGG16 network Simonyan & Zisserman (2015) and a CNN network, respectively. Our numerical evaluations are performed under but not limited to the setting where three clients need to be selected from $K = 10$ total remote clients (more details in App.A.6). Note that the estimated covariance matrices are normalized before further computation. All experiments are repeated with three random seeds.

---

[1]also referred to as Cover's random binning Gamal & Kim (2011) or Gel'fand-Pinsker coding Gel'fand & Pinsker (1980)

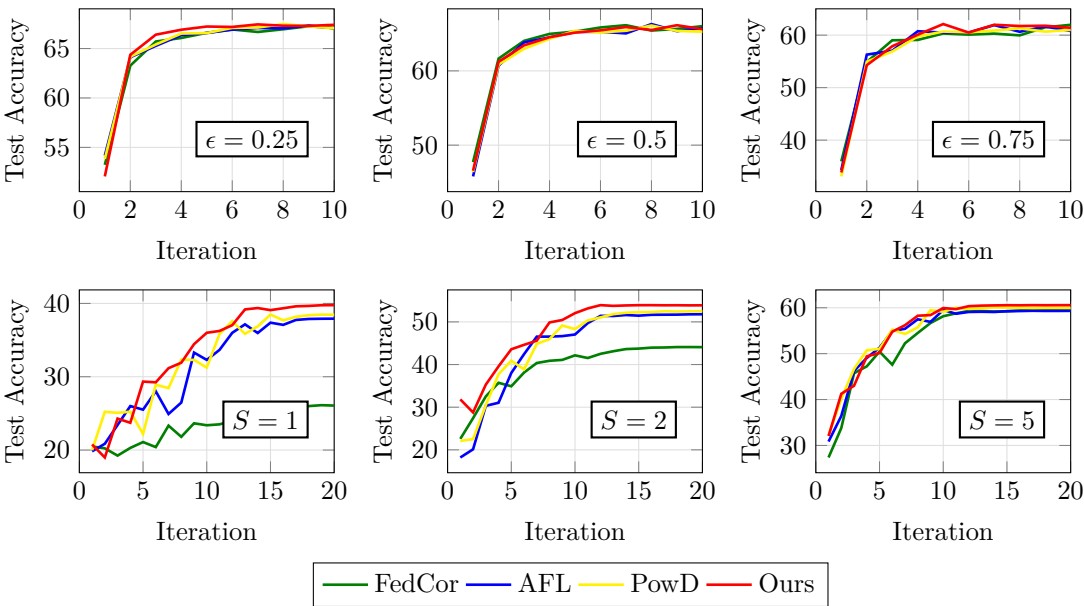

Figure 3: Convergence comparison between our client selection strategy and the SOTA methods on CIFAR-10. Each figure represents one heterogeneity. All methods in one figure share the same hyperparameters except for the client selection strategy.

Table 1: Test accuracy (mean ± `std`) comparison to SOTA methods on CIFAR-10 and Fashion-MNIST.

| **CIFAR-10** | Partition by Shards | | | Partition by Bias Level | | |
|---|---|---|---|---|---|---|
| | $S = 1$ | $S = 2$ | $S = 5$ | $\epsilon = 0.25$ | $\epsilon = 0.5$ | $\epsilon = 0.75$ |
| FedCor | $27.68 \pm 1.52$ | $44.14 \pm 5.23$ | $59.68 \pm 0.67$ | $\mathbf{67.55} \pm 0.37$ | $\mathbf{66.46} \pm 0.52$ | $62.39 \pm 0.69$ |
| AFL | $38.23 \pm 0.87$ | $51.92 \pm 0.58$ | $59.65 \pm 1.13$ | $\mathbf{67.55} \pm 0.10$ | $66.14 \pm 0.36$ | $62.18 \pm 0.31$ |
| PowD | $38.83 \pm 1.99$ | $52.53 \pm 0.71$ | $\mathbf{60.77} \pm 0.46$ | $67.52 \pm 0.21$ | $65.99 \pm 0.36$ | $62.06 \pm 1.42$ |
| Ours | $\mathbf{40.19} \pm 0.84$ | $\mathbf{54.06} \pm 0.44$ | $60.64 \pm 1.53$ | $67.52 \pm 0.08$ | $66.30 \pm 0.71$ | $\mathbf{62.84} \pm 0.56$ |
| **Fashion-MNIST** | Partition by Shards | | | Partition by Bias Level | | |
| | $S = 1$ | $S = 2$ | $S = 5$ | $\epsilon = 0.25$ | $\epsilon = 0.5$ | $\epsilon = 0.75$ |
| FedCor | $20.15 \pm 2.82$ | $31.18 \pm 4.81$ | $42.11 \pm 8.16$ | $71.70 \pm 0.21$ | $68.92 \pm 2.92$ | $64.76 \pm 2.37$ |
| AFL | $20.27 \pm 8.27$ | $37.33 \pm 3.22$ | $48.93 \pm 4.77$ | $71.23 \pm 0.40$ | $69.92 \pm 0.29$ | $65.27 \pm 3.53$ |
| PowD | $\mathbf{24.13} \pm 11.49$ | $44.29 \pm 3.26$ | $52.78 \pm 5.33$ | $\mathbf{71.85} \pm 0.46$ | $70.33 \pm 0.46$ | $64.47 \pm 2.34$ |
| Ours | $20.08 \pm 2.08$ | $\mathbf{44.60} \pm 6.17$ | $\mathbf{59.80} \pm 4.25$ | $\mathbf{71.85} \pm 0.15$ | $\mathbf{70.64} \pm 0.69$ | $\mathbf{66.98} \pm 2.56$ |

To evaluate the effect of gradient correlation on the performance of `GCFed` and the relevant baselines in heterogeneous FL settings, we consider two dataset partitioning methods and three settings for each method as follows

**Partition by Shards (PS)**: The dataset is divided into $K \cdot S = 10 \cdot S$ total shards, where $S$ is the hyperparameter that controls the heterogeneity level. Within each shard, all the data share the same label. We test $S = 1, 2$ and $5$.

**Partition by Bias Level (PB)**: Each client is randomly assigned a favorite/biased label. With the same bias level $\epsilon \in [0, 1]$ for each client, $\epsilon \cdot |\mathcal{D}_k|$ data points of each local dataset are drawn according to the favorite label from the entire dataset, with the rest drawn uniformly from the remaining dataset. A greater $\epsilon$ value indicates more heterogeneity. We test $\epsilon = 0.25, 0.5$ and $0.75$.

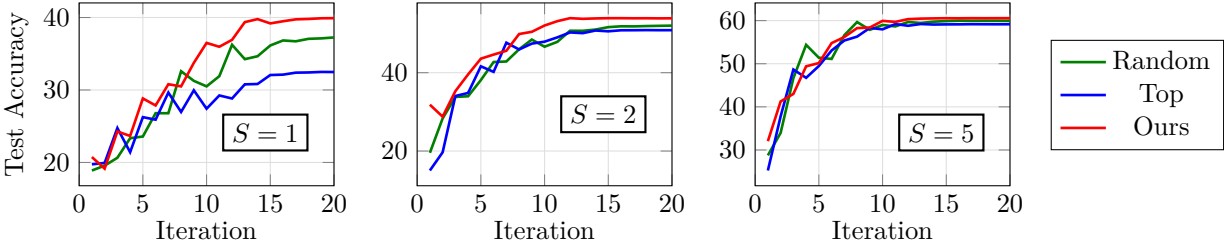

Figure 4: Ablation study on layer-wise client selection implementations on CIFAR-10 for $S = 1, 2$ and 5 under PS settings.

Table 2: Ablation study of layer-wise client selection.

| CIFAR-10 | $S = 1$ | $S = 2$ | $S = 5$ | $\epsilon = 0.25$ | $\epsilon = 0.5$ | $\epsilon = 0.75$ |
|---|---|---|---|---|---|---|
| Random | 37.25 | 52.02 | 59.95 | **67.68** | 65.91 | 62.17 |
| Top | 32.48 | 50.86 | 59.16 | 67.40 | 65.52 | 61.54 |
| Ours | **40.19** | **54.06** | **60.64** | 67.52 | **66.30** | **62.84** |
| **Fashion-MNIST** | $S = 1$ | $S = 2$ | $S = 5$ | $\epsilon = 0.25$ | $\epsilon = 0.5$ | $\epsilon = 0.75$ |
| Random | 19.20 | 36.98 | 51.26 | 71.71 | 70.29 | 65.60 |
| Top | 17.59 | 40.07 | 53.43 | 71.34 | 68.98 | 62.85 |
| Ours | **20.08** | **44.60** | **59.80** | **71.85** | **70.64** | **66.98** |

## 5.2 Compare to SOTA Client Selection Methods

To show the performance of our client selection strategy, we compare it to three baselines: FedCor Tang et al. (2022), AFL Goetz et al. (2019) and POWER-OF-CHOICE (PowD) Jee Cho et al. (2022). All three baselines perform selection based on clients' local losses, with FedCor additionally considering the correlation between local losses. We illustrate the convergence process and the final test accuracy of the converged models. Tab.1 shows that our method achieves the best test accuracy among 8 out of 12 settings. In 3 out of the losing cases, our method lags behind the best by small margins, no more than 0.16%. While in the winning cases, our method leads the second-best by up to 1.83% on CIFAR-10 and 7.02% on Fashion-MNIST. In Fig.3, our method demonstrates the fastest and the most stable convergence compared to the baselines. Particularly, under the PS settings, where the local dataset is more heterogeneous, our method outperforms the baselines significantly.

## 5.3 Layer-wise Client Selection Ablation Study

We conduct ablation studies on layer-wise client selection methods to verify that the superiority of our strategy does not solely result from the layer-wise application. Two ablation implementations are used for comparison. "Random" refers to select clients randomly without considering the estimated covariance matrix for each layer. "Top" refers to using the estimated covariance matrix, but ignoring the correlation, selecting the clients with the largest values along the diagonal of the covariance matrix layer-wise. Other settings are kept the same for all three implementations. As shown in Fig.4 and Tab.2, our method dominates the others under various settings. The results strengthen our argument that exploiting the clients' gradient correlation improves convergence.

## 5.4 Rate Allocation Gains

In Fig.5, we present the information-theoretic gains that ideal vector quantizers could achieve by adopting our optimal rate allocation compared with the counterpart that neglects the gradient correlation. In this paper, we aim to derive the optimal rate allocation and verify its effectiveness. Please refer to Zamir & Feder (1996); Yang et al. (2008) for the practical designs approaching the theoretical limits.

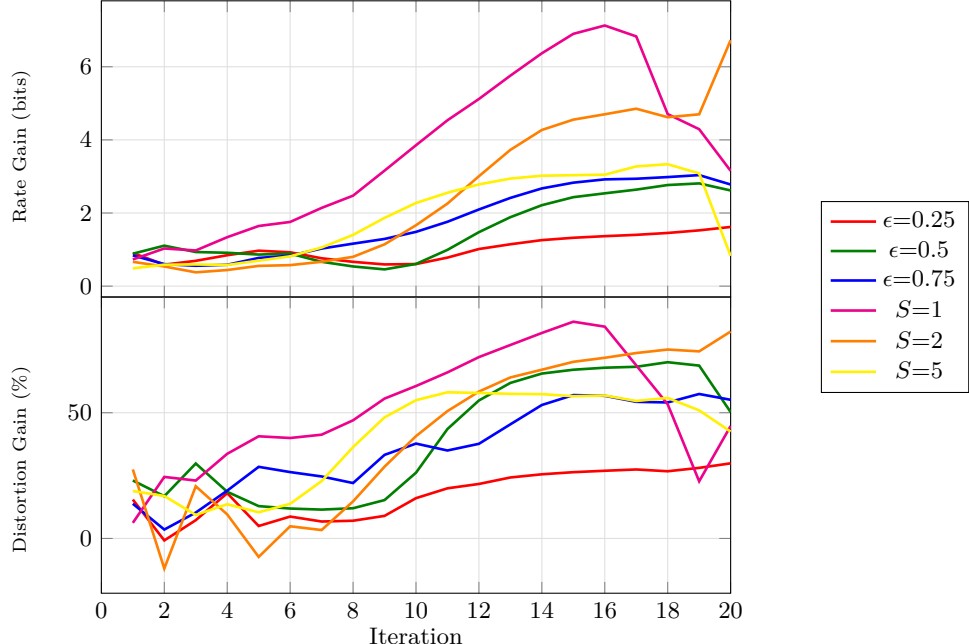

Figure 5: Information-theoretic gains of our correlation-aware optimal rate allocation.

Table 3: Combined performance comparison on CIFAR-10.

| Methods | $S=1$ | $S=2$ | $S=5$ | $\epsilon$=0.25 | $\epsilon$=0.5 | $\epsilon$=0.75 |
|---|---|---|---|---|---|---|
| Corr-agnostic | 24.55 | 42.93 | 56.05 | 67.63 | 65.17 | 63.03 |
| Corr-aware | 26.77 | 49.95 | 60.84 | 67.77 | 65.57 | 63.46 |
| Centralized | 39.63 | 52.71 | 61.06 | 67.93 | 65.67 | 64.11 |

In Fig.5, the top panel shows the gain by the number of bits under the same distortion constraint. The bottom panel shows the gain in terms of the percentage of distortion decrease under the same sum-rate constraint. The simulations validate our technical results, showing consistent gains when comparing "correlation-aware" with "correlation-agnostic" rate allocation throughout the entire training process under all heterogeneity settings. Specifically, the gains become larger in the later iteration rounds as the model converges and correlation increases. Furthermore, we observe more potential gains when local data is more heterogeneous.

## 5.5 Client Selection + Rate Allocation

The above sections demonstrate the promising performance of our two proposed approaches when applied individually. Now, we combine them and evaluate the joint performance – see Tab.3. Following the same client selection strategy (the one we proposed), we evaluate our method against two variants with different rate allocation schemes. "Corr-aware" and "Corr-agnostic" refer to the rate allocation schemes that consider/neglect the gradient correlation as described in Sec.4.3, respectively. "Centralized" represents the performance upper bound obtained when all local datasets are available at the PS. The distortion in this case results from the rate constraint between the PS and this particular client, whose rate constraint is the same as the sum-rate in the other two implementations. Tab.3 shows that one may achieve a significant performance boost by utilizing our proposed correlation-aware optimal rate allocation and even approach the ideal case in some settings.

# 6    Conclusion

This paper introduces a novel framework that exploits the correlation between clients' local gradients to reduce communication cost and improve convergence in heterogeneous FL. Specifically, we propose an information-theoretic formulation, solve the associated optimization problem using convex semidefinite relaxation techniques, and leverage the solution to guide practical implementations for client selection and rate allocation. Extensive experimental results validate the effectiveness of our proposed approach across different FL scenarios.

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

## A  Appendix

### A.1  Client Selection Strategy Details

Here, we describe our client selection strategy in matrix formulation. For simplicity, we consider the scenario in which two out of three clients must be selected. The general case follows in a rather straightforward manner from this example.

Let us begin by considering the covariance matrix $\Sigma_G \in \mathbb{R}^{3 \times 3}$ as

$$\Sigma_G = \begin{bmatrix} \mathrm{var}(G_1) & \mathrm{cov}(G_1, G_2) & \mathrm{cov}(G_1, G_3) \\ \mathrm{cov}(G_2, G_1) & \mathrm{var}(G_2) & \mathrm{cov}(G_2, G_3) \\ \mathrm{cov}(G_3, G_1) & \mathrm{cov}(G_3, G_2) & \mathrm{var}(G_3) \end{bmatrix} \tag{10}$$

As described in SEC. 4.2, first, we construct a "combination matrix", $C$, as follows. The first $K - 1 = 2$ rows of $C$ are each a boolean vector, representing one of the $\binom{3}{2}$ combinations of choosing 2 out of 3 elements. The last row is the all-one vector. Accordingly

$$C = \begin{bmatrix} 1 & 1 & 0 \\ 1 & 0 & 1 \\ 0 & 1 & 1 \\ 1 & 1 & 1 \end{bmatrix} \tag{11}$$

Left and right multiplying $\Sigma_G$ with $C$ and $C^T$, we define a matrix $Q \in \mathbb{R}^{4 \times 4}$ as

$$Q \triangleq C \Sigma_G C^T = \begin{bmatrix} 1 & 1 & 0 \\ 1 & 0 & 1 \\ 0 & 1 & 1 \\ 1 & 1 & 1 \end{bmatrix} \begin{bmatrix} \mathrm{var}(G_1) & \mathrm{cov}(G_1, G_2) & \mathrm{cov}(G_1, G_3) \\ \mathrm{cov}(G_2, G_1) & \mathrm{var}(G_2) & \mathrm{cov}(G_2, G_3) \\ \mathrm{cov}(G_3, G_1) & \mathrm{cov}(G_3, G_2) & \mathrm{var}(G_3) \end{bmatrix} \begin{bmatrix} 1 & 1 & 0 & 1 \\ 1 & 0 & 1 & 1 \\ 0 & 1 & 1 & 1 \end{bmatrix} \tag{12}$$

Note now that the diagonal elements of $Q$ represent the variance of sums as

$$\mathrm{diag}(Q) = (\mathrm{var}(G_1 + G_2), \mathrm{var}(G_1 + G_3), \mathrm{var}(G_2 + G_3), \mathrm{var}(G_1 + G_2 + G_3)) \tag{13}$$

Then, we construct another "subtracting matrix" $S \in \mathbb{R}^{3 \times 4}$ as

$$S = \begin{bmatrix} 1 & 0 & 0 & -1 \\ 0 & 1 & 0 & -1 \\ 0 & 0 & 1 & -1 \end{bmatrix} \tag{14}$$

Left and right multiplying $Q$ with $S$ and $S^T$, result in a matrix $P \in \mathbb{R}^{3 \times 3}$ with diagonal elements being the expected distortion between (i) each of the "choosing two out of three" choices. and (ii) the choice of having all three clients active:

$$P = SQS^T, \tag{15}$$

$$\begin{aligned} \mathrm{diag}(P) = (&\mathrm{var}(G_1 + G_2) - \mathrm{var}(G_1 + G_2 + G_3), \mathrm{var}(G_1 + G_3) - \mathrm{var}(G_1 + G_2 + G_3), \\ &\mathrm{var}(G_2 + G_3) - \mathrm{var}(G_1 + G_2 + G_3)) \end{aligned} \tag{16}$$

Finally, the smallest absolute value among $\mathrm{diag}(P)$ indicates the optimal client selection. Note that one can directly evaluate $P$ as $P = (S \cdot C)Q(S \cdot C)^T$.

### A.2  Further details on auxiliary variables in Sec. 4.3

Given the source sequences $G_1^n, G_2^n, ..., G_K^n$ and the unique encoder at each client $k$, $f_k : \mathbb{R}^n \to \{1, 2, ..., M_k\}, k \in [K]$, we denote the output of the encoders as $W_{\mathcal{K}} = (W_1, W_2 ..., W_K)$ with $W_k = f_k(G_k^n), k \in [K]$. The assumptions stated in Sec.3.3 indicate that for each source sequence $G_k^n$, there are $n$ i.i.d. copies,

i.e., $G_k(i), i \in [n]$ and $G_k(i) \perp G_k(i')$ for $i \neq i'$. We assume that correlations exist between the same indices across different local gradient sequences, meaning $G_k(i)$ and $G_{k'}(i)$ for $k \neq k'$ and $i \in [n]$ are correlated.

Based on the covariance matrix of the source $\Sigma_G$, we define a positive definite covariance matrix $\Sigma_Z$ such that $\Sigma_G^{-1} + \Sigma_Z^{-1}$ is a diagonal matrix. The existence of $\Sigma_Z$ is obvious because any positive definite diagonal matrix $\Delta$ with sufficiently large diagonal entries makes $(\Delta - \Sigma_G^{-1})^{-1}$ a feasible solution to $\Sigma_Z$. And we denote $\mathbb{N}(\Sigma_G)$ as the set of feasible $\Sigma_Z$ given $\Sigma_G$.

Following the definition of $\Sigma_Z$, we let $Z_1^n, Z_2^n, ..., Z_K^n$ be a Gaussian random vector with zero-mean and covariance matrix $\Sigma_Z \in \mathbb{N}(\Sigma_G)$. We assume $(Z_1^n, Z_2^n, ..., Z_K^n)$ is independent of $(G_1^n, G_2^n, ..., G_K^n)$ and define $(Y_1^n, Y_2^n, ..., Y_K^n) = (G_1^n + Z_1^n, G_2^n + Z_2^n, ..., G_K^n + Z_K^n)$. One may view $(Y_1^n, Y_2^n, ..., Y_K^n)$ as a remote source. Due to the Gaussianity of $(Z_1^n, Z_2^n, ..., Z_K^n)$ and its independence with $(G_1^n, G_2^n, ..., G_K^n)$, we know that $(Y_1^n, Y_2^n, ..., Y_K^n)$ is jointly Gaussian distributed with $(G_1^n, G_2^n, ..., G_K^n)$. Since $\Sigma_G^{-1} + \Sigma_Z^{-1}$ is defined as a diagonal matrix, which represents the distortion covariance matrix of the linear MMSE estimation for $(G_1^n, G_2^n, ..., G_K^n)$ given $(Y_1^n, Y_2^n, ..., Y_K^n)$, we obtain that $G_1^n, G_2^n, ..., G_K^n$ are independent conditioned on $(Y_1^n, Y_2^n, ..., Y_K^n)$. To facilitate representations, we shall define $((G_1^n)^T, (G_2^n)^T, ..., (G_K^n)^T)^T$, $((Y_1^n)^T, (Y_2^n)^T, ..., (Y_K^n)^T)^T$, $((Z_1^n)^T, (Z_2^n)^T, ..., (Z_K^n)^T)^T$ as $\mathbf{G}, \mathbf{Y}, \mathbf{Z}$, respectively.

With the above "source"-related variables defined, we introduce a couple of more in terms of MMSE estimation. First, we define a diagonal matrix $\Sigma = \text{diag}(\Sigma_1, \Sigma_2 ..., \Sigma_K)$, with $\Sigma_k, k \in [K]$ being the diagonal entries of $(\Sigma_G^{-1} + \Sigma_Z^{-1})^{-1}$. And we assume that $0 \preceq \text{diag}(\Sigma_1, ..., \Sigma_K) \preceq \Sigma_G$. In fact, $\Sigma_k$ represents the distortion of the MMSE estimation for $G_k^n$ given $\mathbf{Y}$, which is denoted as $\Sigma_k = \Sigma_{G_k^n|\mathbf{Y}}$. Then, we define $D$ to be the the distortion covariance matrix of the MMSE estimation for $\mathbf{G}$ given $W_{\mathcal{K}}$, i.e., $D = \Sigma_{\mathbf{G}|W_{\mathcal{K}}}$. Moreover, we define another diagonal matrix $\Gamma = \text{diag}(\Gamma_1, \Gamma_2 ..., \Gamma_K)$, with diagonal elements $\Gamma_k$ representing the distortion of the MMSE estimation for $G_k^n$ given $\mathbf{Y}$ and $W_k$, i.e., $\Gamma_k = \Sigma_{G_k^n|\mathbf{Y},W_k}$. With the meaning behind all the variables introduced, one can follow the derivation of (10) in Wang et al. (2010) to prove that between $D$ and $\Gamma_k$, the following holds:

$$\text{diag}(\Gamma_1, ..., \Gamma_K) \preceq (D^{-1} + \Sigma_Z^{-1})^{-1}. \tag{17}$$

Following the Theorem 1 and Theorem 2 in Wang et al. (2010), one can derive the lower bound $\underline{R}^*(d)$ and upper bound $\bar{R}^*(d)$ and prove that the two share the same form except for one constraint, which is equation 17. The upper bound requires $\text{diag}(\Gamma_1, ..., \Gamma_K) = (D^{-1} + \Sigma_Z^{-1})^{-1}$, while the lower bound only requires the inequality $\text{diag}(\Gamma_1, ..., \Gamma_K) \preceq (D^{-1} + \Sigma_Z^{-1})^{-1}$.

### A.3 The iterative algorithm to solve for upper and lower bounds

In order to solve equation 7 using numerical convex optimization solvers, it is often required to rewrite the constraint in matrix form. Therefore, we need to rewrite the last constraint. On can do so applying the matrix inversion lemma

$$(D^{-1} + \Sigma_Z^{-1})^{-1} = D - D(\Sigma_Z + D)^{-1}D. \tag{18}$$

By substituting "=" with "$\preceq$", the last constraint of equation 7 is relaxed as follows:

$$0 \preceq D - \text{diag}(\Gamma_1, ..., \Gamma_K) - D(\Sigma_Z + D)^{-1}D, \tag{19}$$

and by Schur complement, this is equivalent to

$$0 \preceq \begin{pmatrix} \Sigma_Z + D & D \\ D & D - \text{diag}(\Gamma_1, ..., \Gamma_K) \end{pmatrix}. \tag{20}$$

The original optimization problem is equation 7 relaxed and converted to the following convex form, which we define as the lower bound:

$$\underline{R}^*(d) \triangleq$$

$$\max_{\Sigma_1,...,\Sigma_K} \min_{D,\Gamma_1,...,\Gamma_K} \frac{1}{2} \log \frac{|\Sigma_G + \Sigma_Z| \, |\text{diag}(\Sigma_1,...,\Sigma_K)|}{|D + \Sigma_Z| \, |\text{diag}(\Gamma_1,...,\Gamma_K)|}$$

$$\text{s.t.} \quad \Sigma_Z = (\text{diag}(\Sigma_1,...,\Sigma_K)^{-1} - \Sigma_G^{-1})^{-1},$$

$$0 \preceq \text{diag}(\Sigma_1,...,\Sigma_K) \preceq \Sigma_G,$$

$$0 \preceq D \preceq \Sigma_G,$$

$$\mathbb{1} \cdot D \cdot \mathbb{1} \le d,$$

$$0 \preceq \text{diag}(\Gamma_1,...,\Gamma_K),$$

$$0 \preceq \begin{pmatrix} \Sigma_Z + D & D \\ D & D - \text{diag}(\Gamma_1,...,\Gamma_K) \end{pmatrix}. \tag{21}$$

Note, $\underline{R}^*(d)$ is not only the lower bound for the Berger-Tung upper bound $\bar{R}^*(d)$, but also the lower bound for the minimum sum-rate $R^*(d)$ Wang et al. (2010); Wang & Chen (2014). When denoting $\Sigma^* = \text{diag}(\Sigma_1^{*-1},...,\Sigma_K^{*-1})$ as the optimal solution to the max part of the above optimization problem, it can be verified that $(\text{diag}(\Sigma_1^{*-1},...,\Sigma_K^{*-1}) - \Sigma_G^{-1})$ must be rank-deficient, which numerically prevents its inversion to calculate $\Sigma_Z$. Therefore, we need to handle it by projection. Let $\alpha_1,...\alpha_p$ denote the positive eigenvalues of $(\text{diag}(\Sigma_1^{*-1},...,\Sigma_K^{*-1}) - \Sigma_G^{-1})$, and let $\pi_1,...,\pi_p$ denote the corresponding eigenvectors. We define

$$\Lambda \triangleq \text{diag}(\frac{1}{\alpha_1},...,\frac{1}{\alpha_p}) \qquad \text{and} \qquad \Pi \triangleq (\pi_1,...,\pi_p), \tag{22}$$

which satisfies that

$$\Sigma_Z^{-1} = \Pi^T \Lambda^{-1} \Pi \qquad \text{and} \qquad \Sigma_Z = \Pi \Lambda \Pi^T. \tag{23}$$

The projection on the objective function is performed as

$$\frac{1}{2} \log \frac{|\Sigma_G + \Sigma_Z| \, |\text{diag}(\Sigma_1,...,\Sigma_K)|}{|D + \Sigma_Z| \, |\text{diag}(\Gamma_1,...,\Gamma_K)|} = \frac{1}{2} \log \frac{|\Pi^T \Sigma_G \Pi + \Lambda| \, |\text{diag}(\Sigma_1,...,\Sigma_K)|}{|\Pi^T D \Pi + \Lambda| \, |\text{diag}(\Gamma_1,...,\Gamma_K)|}. \tag{24}$$

Performing projection on the last constraint of equation 21, we get

$$0 \preceq \begin{pmatrix} \Pi^T D \Pi + \Lambda & \Pi^T D \\ D \Pi & D - \text{diag}(\Gamma_1,...,\Gamma_K) \end{pmatrix}. \tag{25}$$

The final convex optimization problem that is solvable using numerical convex optimization solvers is as

$$\max_{\Sigma_1,...,\Sigma_K} \min_{D,\Gamma_1,...,\Gamma_K} \frac{1}{2} \log \frac{|\Pi^T \Sigma_G \Pi + \Lambda| \, |\text{diag}(\Sigma_1,...,\Sigma_K)|}{|\Pi^T D \Pi + \Lambda| \, |\text{diag}(\Gamma_1,...,\Gamma_K)|}$$

$$\text{s.t.} \quad 0 \preceq D \preceq \Sigma_G,$$

$$\mathbb{1} \cdot D \cdot \mathbb{1} \le d,$$

$$0 \preceq \text{diag}(\Gamma_1,...,\Gamma_K),$$

$$0 \preceq \begin{pmatrix} \Pi^T D \Pi + \Lambda & \Pi^T D \\ D \Pi & D - \text{diag}(\Gamma_1,...,\Gamma_K) \end{pmatrix}. \tag{26}$$

Now, we introduce our algorithm in detail. As we realize that there is a saddle point in the optimization problem in (26), we develop an iterative algorithm to solve the minimization and maximization parts individually, and gradually reach the saddle point, which is also referred to as the optimal solution.

**Initialization:** Solving the minimization part requires an initial guess of $\Sigma_1,...,\Sigma_K$. In numerical simulations, we found that these values are of vital importance for the algorithm to converge. Thus, instead of

randomly initializing their values, we use Wyner's common information Wyner (1975) to obtain the initial values for $\Sigma_1, ..., \Sigma_K$. First, we introduce a new variable $\Theta$ as

$$(\Sigma_G^{-1} + \Theta^{-1})^{-1} = D. \tag{27}$$

Then the max part of the problem, which is in a concave form, becomes

$$\max_{\Sigma_1, ..., \Sigma_K} \frac{1}{2} \log \frac{|\Sigma_G + \Theta| \, |\text{diag}(\Sigma_1, ..., \Sigma_K)|}{|\text{diag}(\Sigma_1, ..., \Sigma_K) + \Theta| \, |\text{diag}(\Gamma_1, ..., \Gamma_K)|}$$
$$\text{s.t.} \quad 0 \preceq \text{diag}(\Sigma_1, ..., \Sigma_K) \preceq \Sigma_G, \tag{28}$$
$$(\text{diag}(\Gamma_1^{-1}, ..., \Gamma_K^{-1}) - \Theta^{-1})^{-1} \preceq \text{diag}(\Sigma_1, ..., \Sigma_K) \preceq \Sigma_G.$$

If we consider when $d \approx \mathbb{1} \cdot \Sigma_G \cdot \mathbb{1}$, the diagonal entries of $\Theta$ will go to infinity because of the definition $\Theta = (D^{-1} + \Sigma_G^{-1})^{-1}$, causing the result of $|\text{diag}(\Sigma_1, ..., \Sigma_K) + \Theta|$ not depending on $(\Sigma_1, ..., \Sigma_K)$. Furthermore, if we neglect the second constraint for now, we arrive at a simple formulation:

$$\max_{\Sigma_1, ..., \Sigma_K} \frac{1}{2} \log |\text{diag}(\Sigma_1, ..., \Sigma_K)|$$
$$\text{s.t.} \quad 0 \preceq \text{diag}(\Sigma_1, ..., \Sigma_K) \preceq \Sigma_G, \tag{29}$$

which we find to be exactly the optimization problem for determining the Wyner's common information when $K = 2$, i.e., if there are only two random variables ($K = 2$). This is equivalent to

$$\min_{P_{W|G_1, G_2}} I(G_1, G_2; W)$$
$$\text{s.t.} \quad G_1 \longleftrightarrow W \longleftrightarrow G_2 \quad \text{form a Markov Chain.} \tag{30}$$

**Full Algorithm:** The initialization process prepares us to introduce the full algorithm that solves the optimization problem.

Step 1: We start with the optimization in (29) to generate initial guesses for $\Sigma_1, ..., \Sigma_K$. We denote the obtained values as $\Sigma_1^{int}, ..., \Sigma_K^{int}$.

Step 2: We fix the value for $\Sigma_1^{int}, ..., \Sigma_K^{int}$ and solve the minimization part of (26), which is in convex form, as

$$\min_{D, \Gamma_1, ..., \Gamma_K} \frac{1}{2} \log \frac{|\Pi^T \Sigma_G \Pi + \Lambda| \, |\text{diag}(\Sigma_1^{int}, ..., \Sigma_K^{int})|}{|\Pi^T D \Pi + \Lambda| \, |\text{diag}(\Gamma_1, ..., \Gamma_K)|}$$
$$\text{s.t.} \quad 0 \preceq D \preceq \Sigma_G,$$
$$\mathbb{1} \cdot D \cdot \mathbb{1} \leq d, \tag{31}$$
$$0 \preceq \text{diag}(\Gamma_1, ..., \Gamma_K),$$
$$0 \preceq \begin{pmatrix} \Pi^T D \Pi + \Lambda & \Pi^T D \\ D\Pi & D - \text{diag}(\Gamma_1, ..., \Gamma_K) \end{pmatrix},$$

by solving which we will obtain the values for $D^t, \Gamma_1^t, ..., \Gamma_K^t$, where $t$ denotes the number of iterations this algorithm has been applied. Before solving the maximization, we calculate $\Theta^t = (D^{t-1} + \Sigma_G^{-1})^{-1}$.

Step 3: We fix the values for $D^t, \Gamma_1^t, ..., \Gamma_K^t$ and try to solve the maximization part of (26) as

$$\max_{\Sigma_1, ..., \Sigma_K} \frac{1}{2} \log \frac{|\Sigma_G + \Theta^t| \, |\text{diag}(\Sigma_1, ..., \Sigma_K)|}{|\text{diag}(\Sigma_1, ..., \Sigma_K) + \Theta^t| \, |\text{diag}(\Gamma_1, ..., \Gamma_K)|}$$
$$\text{s.t.} \quad 0 \preceq \text{diag}(\Sigma_1, ..., \Sigma_K) \preceq \Sigma_G, \tag{32}$$
$$(\text{diag}(\Gamma_1^{-1}, ..., \Gamma_K^{-1}) - \Theta^{t-1})^{-1} \preceq \text{diag}(\Sigma_1, ..., \Sigma_K) \preceq \Sigma_G.$$

Since the solvers in MATLAB and Python require all the formulations to be linear, which conflicts with the objective function in equation 32, we use Gradient Descent to perform the linearization. By taking the derivative of the objective function in (32) with respect to $\Sigma_1, ..., \Sigma_K$, the problem becomes

$$
\max_{\Sigma_1,...,\Sigma_K} \frac{1}{2} \log |\text{diag}(\Sigma_1, ..., \Sigma_K)| - \frac{1}{2} \text{trace}((\text{diag}(\Sigma_1, ..., \Sigma_K) + \Theta^t)^{-1} \cdot \text{diag}(\Sigma_1, ..., \Sigma_K))
$$
$$
\text{s.t.} \quad 0 \preceq \text{diag}(\Sigma_1, ..., \Sigma_K) \preceq \Sigma_G, \tag{33}
$$
$$
(\text{diag}(\Gamma_1^{-1}, ..., \Gamma_K^{-1}) - \Theta^{t-1})^{-1} \preceq \text{diag}(\Sigma_1, ..., \Sigma_K) \preceq \Sigma_G.
$$

By solving this problem, we obtain a new set of $\Sigma_1^{new}, ..., \Sigma_K^{new}$. We define the gradients as $g_1, ..., g_K = (\Sigma_1^{new} - \Sigma_1), ..., (\Sigma_K^{new} - \Sigma_K)$. By multiplying the gradients by a pre-determined step size and subtracting from the original $\Sigma_1, ..., \Sigma_K$, we perform one step of gradient descent. Keep updating $\Sigma_1, ..., \Sigma_K$ by this method will reach the final values of $\Sigma_1, ..., \Sigma_K$ that maximize the problem (32) with current values of $D^t, \Gamma_1^t, ..., \Gamma_K^t, \Theta^t$. Therefore, by adopting "Successive Convex Optimization" and "Gradient Descent", Step 3 can be implemented as follows:

Step 3.1: Start with a feasible point, which is $\Sigma_1^{int}, ..., \Sigma_K^{int}$.

Step 3.2: Take the derivative of the objective function of equation 32 around the current $\Sigma_1, ..., \Sigma_K$ values, resulting in the objective function of equation 33.

Step 3.3: Solve the linearized optimization problem and calculate the gradients $g_1, ..., g_K$.

Step 3.4: Update $\Sigma_1, ..., \Sigma_K$ with gradients multiplied by the step-size to form the new $\Sigma_1, ..., \Sigma_K$.

Step 3.5: Repeat until convergence. Note that there is a saddle point effect in our max-min problem. The convergence condition is either (i) the objective function values of minimization and maximization being very close or (ii) the gap between them becomes zero or being held at a constant. Until one of these conditions is met, we conclude that we have finished the maximization part (Step 3).

Step 4: $t = t + 1$, and repeat Step 2 and Step 3 until convergence. The convergence condition for Step 4 is defined as either (i) the upper bound and the lower bound of the sum-rate coinciding or (ii) the gap between them becomes zero or being held at a constant.

The lower bound of the sum-rate is

$$
\underline{R}^*(d) = \frac{1}{2} \log \frac{|\Sigma_G + \Sigma_Z| \, |\text{diag}(\Sigma_1, ..., \Sigma_K)|}{|D + \Sigma_z| \, |\text{diag}(\Gamma_1, ..., \Gamma_K)|} \quad , \text{where } D \text{ is the solution to equation 31.} \tag{34}
$$

Substituting the optimal $\Gamma_1, ..., \Gamma_k$ and $\Sigma_Z$ (obtained from solving equation 31) back, the upper bound of the sum-rate follows

$$
\bar{R}^*(d) = \frac{1}{2} \log \frac{|\Sigma_G + \Sigma_Z| \, |\text{diag}(\Sigma_1, ..., \Sigma_K)|}{|D^* + \Sigma_z| \, |\text{diag}(\Gamma_1, ..., \Gamma_K)|} \quad , \text{where } D^* = ((\text{diag}(\Gamma_1, ..., \Gamma_K)^{-1}) - \Sigma_Z^{-1})^{-1}. \tag{35}
$$

The gap between them is simply calculated as

$$
\bar{R}^*(d) - \underline{R}^*(d) = \frac{1}{2} \log \frac{|\Sigma_G + \Sigma_Z| \, |\text{diag}(\Sigma_1, ..., \Sigma_K)|}{|D^* + \Sigma_z| \, |\text{diag}(\Gamma_1, ..., \Gamma_K)|} - \frac{1}{2} \log \frac{|\Sigma_G + \Sigma_Z| \, |\text{diag}(\Sigma_1, ..., \Sigma_K)|}{|D + \Sigma_z| \, |\text{diag}(\Gamma_1, ..., \Gamma_K)|}. \tag{36}
$$

In conclusion, our proposed iterative algorithm obtains the initial values of two fundamental variables by first solving a relaxed problem. Then, by iteratively solving the minimization problem (convex) and solving the maximization problem (concave), we compute the saddle point of the max-min problem. We highlight that executing our algorithm once yields both upper and lower bound values for the minimum sum-rate $R^*(d)$, which is enable by the convex semidefinite relaxation.

## A.4 Generalizations of the problem formulation and results

In the main context, we employ layer-wise rate-distortion analysis, which means we use our iterative algorithm to solve for the corresponding equation 7 under a specific $d$ distortion level for each layer. In the following,

we present three generalized usage of our proposed approaches. We highlight that these generalizations can be easily combined.

**Generalization 1:** Our problem formulation can be seamlessly generalized to multi-layer rate-distortion analysis. Without the loss of generality, let us consider a two-layer regime. Assuming two covariance matrices $\Sigma_G^{(1)}$ and $\Sigma_G^{(2)}$ correspond to the correlation estimation on two layers and each layer has $N^{(1)}$ and $N^{(2)}$ weights. Define the total distortion constraint $d_{total}$ for two layers combined. Our problem formulation is generalized to a joint optimization problems as

$$
\max_{\substack{\Sigma_1^{(1)},...,\Sigma_K^{(1)} \, D^{(1)},\Gamma_1^{(1)},...,\Gamma_K^{(1)} \\ \Sigma_1^{(2)},...,\Sigma_K^{(2)} \, D^{(2)},\Gamma_1^{(2)},...,\Gamma_K^{(2)}}} \min \frac{1}{2}\log \frac{|\Sigma_G^{(1)}+\Sigma_Z^{(1)}| \, |\text{diag}(\Sigma_1^{(1)},...,\Sigma_K^{(1)})|}{|D^{(1)}+\Sigma_Z^{(1)}| \, |\text{diag}(\Gamma_1^{(1)},...,\Gamma_K^{(1)})|} + \frac{1}{2}\log \frac{|\Sigma_G^{(2)}+\Sigma_Z^{(2)}| \, |\text{diag}(\Sigma_1^{(2)},...,\Sigma_K^{(2)})|}{|D^{(2)}+\Sigma_Z^{(2)}| \, |\text{diag}(\Gamma_1^{(2)},...,\Gamma_K^{(2)})|}
$$

$$
\begin{aligned}
\text{s.t.} \quad & 0 \preceq \text{diag}(\Sigma_1^{(1)},...,\Sigma_K^{(1)}) \preceq \Sigma_G^{(1)}, \quad 0 \preceq \text{diag}(\Sigma_1^{(2)},...,\Sigma_K^{(2)}) \preceq \Sigma_G^{(2)}, \\
& 0 \preceq D^{(1)} \preceq \Sigma_G^{(1)}, \quad 0 \preceq D^{(2)} \preceq \Sigma_G^{(2)}, \\
& \frac{N^{(1)}}{N^{(1)}+N^{(2)}} \mathbb{1} \cdot D^{(1)} \cdot \mathbb{1} + \frac{N^{(2)}}{N^{(1)}+N^{(2)}} \mathbb{1} \cdot D^{(2)} \cdot \mathbb{1}^T \le d_{total}, \\
& 0 \preceq \text{diag}(\Gamma_1^{(1)},...,\Gamma_K^{(1)}) \preceq \text{diag}(\Sigma_1^{(1)},...,\Sigma_K^{(1)}), \quad 0 \preceq \text{diag}(\Gamma_1^{(2)},...,\Gamma_K^{(2)}) \preceq \text{diag}(\Sigma_1^{(2)},...,\Sigma_K^{(2)}), \\
& \text{diag}(\Gamma_1^{(1)},...,\Gamma_K^{(1)}) \preceq (D^{(1)^{-1}}+\Sigma_Z^{(1)^{-1}})^{-1}, \quad \text{diag}(\Gamma_1^{(2)},...,\Gamma_K^{(2)}) \preceq (D^{(2)^{-1}}+\Sigma_Z^{(2)^{-1}})^{-1}.
\end{aligned}
\tag{37}
$$

Note that our iterative algorithm is applicable to solve this joint optimization problem.

**Generalization 2:** Another generalization concerns the aggregation function. In the main context and in equation 37, we take FedAvg with an equal amount of local data as an example. Under this setting, we need to left- and right-multiply the defined distortion matrix $D$ by the all-ones vector $\mathbb{1}$ and $\mathbb{1}^T$. We highlight that our formulations can be generalized to all linear aggregation functions. The adjustments lie in the all-ones vector.

Here, we take FedAvg with a different amount of local data as an example. Since each client's contribution to the total distortion is proportional to the size of the local datasets, we define $\mathbf{d} = (\sqrt{\mathbf{d}_1/\sum \mathbf{d}_k}, \sqrt{\mathbf{d}_2/\sum \mathbf{d}_k},...,\sqrt{\mathbf{d}_K/\sum \mathbf{d}_k}) \in \mathbb{R}^K$, where $\mathbf{d}_k$ represents the size of client $k$'s local dataset and $\sum \mathbf{d}_k$ represents the total size of all clients' local data. Replacing $\mathbb{1}$ with $\mathbf{d}$, i.e., $\mathbf{d} \cdot D \cdot \mathbf{d}^T$, yields the distortion constraint for FedAvg with a different amount of local data.

**Generalization 3:** As stated in Sec. 3.3, we model each layer of the gradient at each client as a random variable, and each element/position of the gradient sequence is i.i.d. distributed. Our methodologies can indeed be generalized to random vector cases. This section briefly describes the changes in the rate allocation formulations.

When modeling each layer of the gradient with random vectors, the notation of $\text{diag}(\cdot)$ no longer represents a diagonal matrix. Instead, it represents a symmetric matrix in block-diagonal form. In addition, the newly introduced variables $\Sigma_k$ and $\Gamma_k$, for $k \in [K]$ each represents a PSD matrix. Thus, $\text{diag}(\Sigma_1,...,\Sigma_K)$ and $\text{diag}(\Gamma_1,...,\Gamma_K)$ are block-diagonal matrices with $\Sigma_k$ and $\Gamma_k$ aligning along the diagonal from $k = 1$ to $k = K$. Note that the same algorithm we proposed could be used to numerically solve this generalized formulation.

### A.5   Rate Allocation and Quantization Simulation

In (9), we show the general expression for rate allocation when "binning" is applied. Here, we present more details. First, we present the two extreme cases of $M_k, k \in [K]$ as

$$
M_0 = \text{diag}(\Gamma_1,...\Gamma_K) \quad \text{and} \quad M_K = \text{diag}(\Sigma_1,...,\Sigma_K). \tag{38}
$$

Note that the above expressions are two extreme cases for (8). In later expressions, we use the definitions of $W_k, k \in [K]$ and $\mathbf{Y}$ in Appendix A.2. Then, by defining that $\text{diag}(\Sigma_1,...,\Sigma_K) = (\Sigma_G^{-1} + \Sigma_Z^{-1})^{-1}$ and

$\text{diag}(\Gamma_1, ..., \Gamma_K) = (D^{-1} + \Sigma_Z^{-1})^{-1}$, the optimal rate allocation follows the following equations and they satisfy $\bar{R}^*(d) = \sum_{k \in [K]} R_k$:

$$
\begin{aligned}
R_K &= I(G_K; W_K) \\
&= H(\mathbf{Y}) - H(\mathbf{Y}|W_K) + I(G_K; W_K|\mathbf{Y}) \\
&= \frac{1}{2} \log \frac{|\Sigma_Z(\Sigma_Z - M_K)^{-1}\Sigma_Z|}{|\Sigma_Z(\Sigma_Z - M_{K-1})^{-1}\Sigma_Z|} + \frac{1}{2} \log \frac{|\Sigma_K|}{|\Gamma_K|}, \quad \text{and} \\
R_k &= I(G_k; W_k|W_{k+1}, ..., W_K) \\
&= H(\mathbf{Y}|W_{k+1}, ..., W_K) - H(\mathbf{Y}|W_1, ..., W_K) + I(G_k; W_k|\mathbf{Y}) \\
&= \frac{1}{2} \log \frac{|\Sigma_Z(\Sigma_Z - M_k)^{-1}\Sigma_Z|}{|\Sigma_Z(\Sigma_Z - M_{k-1})^{-1}\Sigma_Z|} + \frac{1}{2} \log \frac{|\Sigma_k|}{|\Gamma_k|}, \quad \text{and} \\
R_1 &= I(G_1; W_1|W_2, ..., W_K) \\
&= H(\mathbf{Y}|W_2, ..., W_K) - H(\mathbf{Y}|W_1, ..., W_K) + I(G_1; W_1|\mathbf{Y}) \\
&= \frac{1}{2} \log \frac{|\Sigma_Z(\Sigma_Z - M_1)^{-1}\Sigma_Z|}{|\Sigma_Z(\Sigma_Z - M_0)^{-1}\Sigma_Z|} + \frac{1}{2} \log \frac{|\Sigma_1|}{|\Gamma_1|}.
\end{aligned}
\tag{39}
$$

As mentioned in Section 4.3, equation 39 and equation 9 represent one of the K! orders, which indicates that we determine client$_K$'s rate by itself first. Then, using client$_K$'s information to determine client$_{K-1}$'s rate, and so on until client$_1$. One can easily show that there are $K!$ permutations with respect to the determining order. We note that these $K!$ orders, as corner points, define a dominant face. Any point lying in this dominant face represents an optimal rate allocation scheme.

When projection is applied due to the rank deficiency of $\Sigma_Z$, we can rewrite the rate allocation as follows:

$$
\frac{1}{2} \log \frac{|\Sigma_Z(\Sigma_Z - M_K)^{-1}\Sigma_Z|}{|\Sigma_Z(\Sigma_Z - M_{K-1})^{-1}\Sigma_Z|} = \frac{1}{2} \log \frac{|\Lambda(\Lambda - \Pi^T M_K \Pi)^{-1}\Lambda|}{|\Lambda(\Lambda - \Pi^T M_{K-1}\Pi)^{-1}\Lambda|}.
\tag{40}
$$

**Correlation-agnostic Rate Allocation:** Following the expressions of rate allocation when "binning" is applied, we present, under the same sum-rate constraint, the individual rate for each client in a contrasting case, meaning that we do not consider the gradient correlation (each client quantizes based on its own data). In this case, only the diagonals of $\Sigma_G$ are known. The individual rate is as follows:

$$
\begin{aligned}
R_k &= I(G_k; U_k) = H(G_k) - H(G_k|W_k) \\
&= \frac{1}{2} \log(|\Sigma_{Gk}||\Sigma_{Gk}^{-1} + \Gamma_k^{-1} - \Sigma_k^{-1}|), \qquad \text{for } k \in [K],
\end{aligned}
\tag{41}
$$

where $\Sigma_{Gk}$ is the $k$-th diagonal element of $\Sigma_G$. Note, this rate allocation still satisfies the same distortion constraint as in equation 7.

**Simulating the quantization process:** With the rate allocation for both correlation-aware and correlation-agnostic rate allocation schemes introduced, we are ready to explain how we simulate the quantization under these rates according to Zamir & Feder (1996); Yang et al. (2008). As shown in the reference, the simulation process for dithered quantization includes 1) adding uniformly distributed noise to the source; 2) reconstructing the sources using linear minimum mean square error (MMSE) estimation.

Step 1: for both binning and no-binning cases, once the rate $R_k$ is determined, because we model each layer of gradients as a single variable, the variance of the noise needs to be added to the source can be calculated as (reduced to a scalar case)

$$
R_k = \frac{1}{2} \log \frac{\Sigma_{Gk}^2 + \sigma_{Nk}^2}{\sigma_{Nk}^2}, \quad k \in [K].
\tag{42}
$$

After obtaining the noise variance $\sigma_{Nk}^2$, $k \in [K]$, we uniformly sample a tensor of noise following $N(0, \sigma_N^2)$ and add it onto the corresponding layer of gradients, resulting in $\hat{G}_1, ..., \hat{G}_K$, which denotes the noisy version of the sources with additive Gaussian noise.

Step 2: for reconstructing the source with linear MMSE estimation, we need to first define

$$\Psi \triangleq (D^{-1} - \Sigma_G^{-1})^{-1}, \tag{43}$$

and then define

$$A \triangleq \Sigma_G \cdot (\Sigma_G + \Psi)^{-1}. \tag{44}$$

To get the reconstructions $\tilde{G}_1, ..., \tilde{G}_K$, we do the follows:

$$[\tilde{G}_1, ..., \tilde{G}_K]^T = A \cdot [\hat{G}_1, ..., \hat{G}_K]^T. \tag{45}$$

Note that this process Zamir & Feder (1996): from the original source $(G_1, ..., G_K)$ to the noisy version $(\hat{G}_1, ..., \hat{G}_K)$ and finally to the denoised version $\tilde{G}_1, ..., \tilde{G}_K$, well-approximates the encode-decode process using a dithered lattice quantizer following the rate allocation $R_k$, for $k \in [K]$. For the "Centralized" case mentioned in Section 5.5, we hypothesize that there is one external client with access to all local clients' gradients, meaning that the external client has direct access to the true FedAvg result. Considering a communication rate constraint between this external client and the server, we still must use the aforementioned steps to simulate the compression and decompression process. The only difference is that all the sum-rate is assigned to the external client without the need for rate allocation under this centralizing setting.

## A.6 Experiment Setting Details

When conducting experiments on CIFAR-10 with VGG16 network, we initialize the network using a pre-trained weights obtained from ImageNet-1K Deng et al. (2009) except for the last output layer for the purpose of efficiently analyzing the effectiveness of our methods. During the training process, we freeze the feature part of the network except for the last three CONV layers and the subsequent ReLU and pooling layers. Also, we unfreeze the entire classifier part, which consists of 3 Fully-Connected(FC) layers, as well as and Dropout layers. When experimenting on Fashion-MNIST, we adopt a simple CNN network with 2 CONV layers and 2 FC layers. All the correlation estimation and related applications are deployed on the 3 FC layers because we focus on the spatial correlation between the same positions in FC layers. For each layer, the server randomly samples 100 indices regardless of the size of the layer. Note that the application of our approach is not limited to our chosen dataset or network architecture.

In terms of FL settings, each client performs 3 rounds of local updates before they communicate with the server, and the total number of communication iterations is set to 20. We use SGD as the optimizer with learning rate 0.03 for i) Partition by shards and 0.01 for ii) Partition by bias level, respectively, with momentum equal to 0.5 and a learning rate decay equal to 0.995. We apply FedAvg to aggregate the clients' updates. We choose the distortion to be $d = 0.1 \cdot (\mathbb{1} \cdot \Sigma_G \cdot \mathbb{1})$ (0.1 times the maximum distortion), which uniquely determines the sum rate.

