# OpenReview forum: "GCFed: Exploiting Gradient Correlation for Client Selection and Rate Allocation in Federated Learning"
_TMLR — Withdrawn by Authors_

### Review · Reviewer_Nsqn · 2025-06-14

**Summary Of Contributions:**

The authors consider a rate-distortion based information-theoretic framework to model a single communication round in Federated Learning (FL). By drawing on classical Berger-Tung results for multi-terminal source coding, they derive bounds on the achievable sum-rate for client gradients. The authors use this formulation to develop client selection and rate allocation strategies that aim to improve communication efficiency.
While the formulation is novel and provides a new connection between FL and information theory, the main theoretical results follow as a direct consequence of theorems in source coding. The assumptions made in the setup, especially A2 through A5, seem to significantly restrict the applicability of the proposed approach. Furthermore, the experimental validation is limited to small-scale datasets with any confidence bounds.
I believe this work is better suited for an information theory-focused venue, such as the IEEE Transactions on Information Theory or ISIT, rather than TMLR.

**Audience:**

Yes

**Claims And Evidence:**

Yes

**Requested Changes:**

The paper would benefit from one of the following two strategies:

(1) Reposition as an information theory contribution. Clearly frame the work as a novel application of multi-terminal source coding to FL, without claiming general practicality. Submit to an information theory venue such as IT or ISIT.

(2) Reinforce empirical validity. Justify assumptions either theoretically (e.g., distributional convergence results) or empirically (e.g., gradient histograms, normality tests). Extend experiments to realistic FL scenarios with large models and real-world client data.

Minor Comments

The notation for $g_t$ and $\hat{g}_t$ is unclear in the rate-distortion definition. The meaning of the source and reconstruction needs to be explicitly defined.

The definition of local gradients in Eq. (2) is ambiguous. The expectation should be clearly specified with respect to the sampling process (e.g., minibatches).

In Algorithm 1, it would be helpful to include concrete values for the number of gradient coordinates used for correlation estimation.
The authors should quantify the overhead introduced by the semidefinite programming step on the server.

References:

Jacot, A., Gabriel, F., & Hongler, C. (2018). Neural Tangent Kernel: Convergence and Generalization in Neural Networks. Advances in Neural Information Processing Systems (NeurIPS).


Alistarh, D., Grubic, D., Li, J., Tomioka, R., & Vojnovic, M. (2017). QSGD: Communication-efficient SGD via Gradient Quantization and Encoding. Advances in Neural Information Processing Systems (NeurIPS).


Bernstein, J., Wang, Y. X., Azizzadenesheli, K., & Anandkumar, A. (2018). signSGD: Compressed Optimization for Non-Convex Problems. International Conference on Machine Learning (ICML).


Aji, A. F., & Heafield, K. (2017). Sparse Communication for Distributed Gradient Descent. Conference on Empirical Methods in Natural Language Processing (EMNLP).

**Strengths And Weaknesses:**

Overall, the assumptions made resemble a first-order approximation of neural networks [1]. Below, I outline concerns with each key assumption.

A2: Independence over time This assumption states that gradients from different communication rounds are independent. This is not valid in realistic training regimes using stochastic gradient descent (SGD), where gradients across rounds are typically highly correlated
due to the continuity of optimization trajectory and shared data. Ignoring this structure limits the realism of the rate-distortion model.

A3: IID-ness intra-layer The assumption that gradient coordinates within a layer are i.i.d. is questionable. In modern architectures, neurons and filters often specialize during training, and intra-layer gradients typically exhibit structured correlations (e.g., block patterns in convolutional layers).

A4: Joint Gaussianity across clients This is perhaps the strongest and least defensible assumption. Gradients across clients are influenced by data heterogeneity, training dynamics, and non-linearities in the model. These gradients often exhibit heavy tails, skewness, and multi-modality, none of which are captured by the Gaussian model.

A5: Independence across layers Backpropagation naturally induces cross-layer dependencies  and batch normalization. This assumption reduces the method to a single-layer view.  The authors claim to handle multi-layer deep networks in appendix A4, but their formulation assumes layer-wise independence, which is a clear contradiction.

These assumptions must either be rigorously justified or replaced with empirically grounded approximations. As it stands, the entire theoretical framework applies only in the limit where the model behaves like a linearized kernel, such as under NTK conditions.

The experimental results demonstrate some gains, but the scope is too narrow for real-world validation:
All datasets are small (CIFAR-10, Fashion-MNIST) and networks shallow (VGG-16, small CNN). These settings are known to exhibit near-kernel behavior, which artificially validates the assumptions made.
Only ten clients are used in each experiment, and heterogeneity is simulated via synthetic label bias. There is no evaluation on real federated datasets or device distributions.
No baseline includes modern gradient compression schemes such as QSGD [2], SignSGD [3], or Top-k sparsification[4].
The Gaussianity of gradients is not empirically verified.
Results are averaged over just three seeds with no confidence intervals or statistical tests.
In sum, the experiments are not sufficient to validate the claims for practical FL systems. Larger-scale evaluation on realistic FL settings (e.g., ImageNet with ResNet or EfficientNet) and empirical study of gradient distributions are necessary.

---

### Review · Reviewer_BN7c · 2025-06-24

**Summary Of Contributions:**

This paper proposes GCFed, which leverages gradient correlation per-layer for client selection and rate allocation. At a high level, it selects the client set that mostly matches the full client gradient correlation under some assumptions, and solves a semidefinite programming problem to allocate the rate. Experiments on CIFAR-10 and Fashion-MNIST demonstrate the effectiveness of the proposed approach in terms of higher accuracy, better convergence, and communication cost savings.

**Audience:**

Yes

**Claims And Evidence:**

No

**Requested Changes:**

See weaknesses.

**Strengths And Weaknesses:**

Strengths:
- A theoretically-principled approach for reducing communication cost in federated learning. The principle of using gradient correlation is novel and worth further exploration inspired by this work.
- Solid theoretical analysis about optimal rate allocation.
- Good empirical evidence demonstrates the effectiveness.

Weaknesses:
- Despite being solid, the theoretical analysis lacks some preliminaries and detailed derivations, which hinders understanding and correctness check. Specifically, for client selection, how does summing up pairwise correlation approximate the overall client gradient update correlation? What is the rigorous goal achieved by the subset selection? Is there explicit connection between better correlation error control and faster convergence? For rate allocation, some backgrounds may need to be included, especially necessary background in Wang et al and Wang & Chen, and Berger-Tung upper bound. Otherwise, I got lost when seeing Eqn. (6), where $\Phi$ and $M_k$ are undefined.

- Experiment validation is limited in scale. It is only conducted on CIFAR-10 and Fashion-MNIST, which is relatively small scale and the heterogeneity is artificially created instead of naturally risen. The method needs to be verified on large-scale datasets such as ImageNet. Moreover, there seems no ablation on the "layer-wise" setting - what if we select client globally instead of layer-wise - which is more practical since gradients are usually transmitted (or not transmitted) to the central server as a whole.

---

### Review · Reviewer_FCuZ · 2025-07-13

**Summary Of Contributions:**

The paper proposes a distortion minimization rate allocation framework for aggregating client updates in federated learning. The method is based on several assumptions such as joint Gaussian gradients etc. Experiments are provided to justify the proposed algorithm.

**Audience:**

Yes

**Claims And Evidence:**

Yes

**Requested Changes:**

Improve presentation. Adding complexity analysis and enrich the experiments.

**Strengths And Weaknesses:**

1. Presentation is not satisfactory. Some key idea and components are not explained clearly so it may seen confusing. For example, what are ClientEncode and ServerDecode functions in Algorithm 1? I find it hard to fully understand the algorithm and how the gradients are aggregated. There should be clear formulas for these key steps.

2. The Gaussian assumption seem too idea and less realistic for me. Given the highly heterogeneous data in FL, assuming joint Gaussian of the gradients may be too restrictive and inpractical.

3. Proposed method requires quite a lot extra computations for computing the covariance matrices and client selection through the combination matrix. It should be analyzed theoretically and justified empirically. Without the complexity comparison, it is hard to measure the benefits of the proposed method. Also, the experiments seem small-scale. More experiments on larger datasets and models should be provided to verify the advantages.

---

### Comment · Action_Editor_ASRR · 2025-05-29
**Please remove the latex command in the title**

Because it cannot be rendered outside this page (for example, in the Action Editor Console and Modify Reviewer Assignments webpage).

---

### Comment · Reviewer_FCuZ · 2025-07-01
**Clear math formulation but lacks practicality; complexity analysis missing**

The paper proposes a distortion minimization rate allocation framework for aggregating client updates in federated learning. While the math formulation is clear, I have several concerns:

1. Presentation is not satisfactory. Some key idea and components are not explained clearly so it may seen confusing. For example, what are ClientEncode and ServerDecode functions in Algorithm 1? I find it hard to fully understand the algorithm and how the gradients are aggregated. There should be clear formulas for these key steps.

2. The Gaussian assumption seem too idea and less realistic for me. Given the highly heterogeneous data in FL, assuming joint Gaussian of the gradients may be too restrictive and inpractical.

3. Proposed method requires quite a lot extra computations for computing the covariance matrices and client selection through the combination matrix. It should be analyzed theoretically and justified empirically. Without the complexity comparison, it is hard to measure the benefits of the proposed method. Also, the experiments seem small-scale. More experiments on larger datasets and models should be provided to verify the advantages.

Overall, although the math formulation of the problem is clear, the algorithm does not seem very practical. The presentation can be improved, and the complexity analysis is important but missing.

---

### Note · Authors · 2025-07-22

**Comment:**

After careful consideration of the reviews from, we believe that the current feedback may not be aligned with our vision for the work, and addressing the concerns raised would require a significant redirection that deviates from the original scope and contributions we intended to highlight.

Given this, following the reviewer's suggestion, we feel it would be more productive to revise and submit the manuscript to an information theory-focused venue that is better aligned with the intended framing and technical focus of our work.

We thank the reviewers and editors for their time and consideration, and we sincerely appreciate the opportunity to have our work reviewed at TMLR.

**Withdrawal Confirmation:**

I have read and agree with the venue's withdrawal policy on behalf of myself and my co-authors.